# A Chimeric ORF Fusion Phenotypic Reporter for *Cryptococcus neoformans*

**DOI:** 10.3390/jof10080567

**Published:** 2024-08-12

**Authors:** Louis S. Phillips-Rose, Chendi K. Yu, Nicholas P. West, James A. Fraser

**Affiliations:** Australian Infectious Diseases Research Centre, School of Chemistry & Molecular Biosciences, The University of Queensland, Brisbane, QLD 4072, Australia; l.phillipsrose@uq.edu.au (L.S.P.-R.); chendi.yu@uq.net.au (C.K.Y.);

**Keywords:** *Cryptococcus neoformans*, fungal pathogen, fusion gene, reporter, *amdS*, fluorescent protein

## Abstract

The plethora of genome sequences produced in the postgenomic age has not resolved many of our most pressing biological questions. Correlating gene expression with an interrogatable and easily observable characteristic such as the surrogate phenotype conferred by a reporter gene is a valuable approach to gaining insight into gene function. Many reporters including *lacZ*, *amdS,* and the fluorescent proteins mRuby3 and mNeonGreen have been used across all manners of organisms. Described here is an investigation into the creation of a robust, synthetic, fusion reporter system for *Cryptococcus neoformans* that combines some of the most useful fluorophores available in this system with the versatility of the counter-selectable nature of *amdS*. The reporters generated include multiple composition and orientation variants, all of which were investigated for differences in expression. Evaluation of known promoters from the *TEF1* and *GAL7* genes was undertaken, elucidating novel expression tendencies of these biologically relevant *C. neoformans* regulators of transcription. Smaller than *lacZ* but providing multiple useful surrogate phenotypes for interrogation, the fusion ORF serves as a superior whole-cell assay compared to traditional systems. Ultimately, the work described here bolsters the array of relevant genetic tools that may be employed in furthering manipulation and understanding of the WHO fungal priority group pathogen *C. neoformans*.

## 1. Introduction

Despite the plethora of genome sequences determined in the postgenomic age, many of the most pressing biological questions have yet to be resolved [1,2,3,4,5]. As the then-director of the Broad Institute famously said in his 2003 nanolecture on the human genome sequence, “genome: bought the book; hard to read” [6]. Knowing the sequence or location of a gene is not enough; it is equally important to know when and where genes are turned on or off and their level of transcription [7]. Whilst the phenotypic consequences of transcription are sometimes easily observed, the determination of the transcriptional status of a gene often requires analysis at the molecular level [8,9]. It is therefore extremely useful if gene expression can be correlated with an easily observable, obvious characteristic such as the surrogate phenotype conferred by a reporter gene. The archetypical ideal of a reporter is embodied by the first genetic element to be deeply studied and which remains prevalent as a tool for analysis of gene expression today, the *lacZ* gene of the *lac* operon in *Escherichia coli*.

The *E. coli lac* operon encoding the proteins required for lactose utilisation was the first genetic element shown to be differentially expressed in response to environmental cues [10,11,12]. Expression of the polycistronic transcript encoding LacZ (β-galactosidase), LacY (β-galactoside permease) and LacA (β-galactoside transacetylase) only occurs during insufficiency of the preferred carbon source glucose and in the presence of the substrate lactose. Key to the utility of *lacZ* as a reporter of transcription are chromogenic substrates such as 5-bromo-4-chloro-3-indolyl-β-D-galactopyranoside (X-gal) or ortho-nitrophenyl-β-galactoside (ONPG) [13,14]. When LacZ is produced in the presence of these compounds, it catalyses the production of a blue or yellow colour, respectively, a technology that has advanced with the development of substrates such as fluorescein di-β-D-galactopyranoside (FDG) that fluoresce when hydrolysed by this enzyme [15,16,17]. As a result, *lacZ* has served as an important reporter gene in bacterial and eukaryotic systems for over half a century, providing a highly efficient and deceptively simple approach to revealing alterations in the transcription of genes of interest [18,19,20,21].

The primary use case for *lacZ* is creating a visual phenotype using a chromogenic substrate. However, at 3075 bp, *lacZ* is the 57th largest of the 4530-protein coding ORFs in *E. coli* K12; this large size makes the sequence hard to work with in other species [22]. Furthermore, the reporter phenotype requires the addition of a substrate and is only semiquantitative (intensity of blue colour) or requires a cell lysate with a corresponding biochemical assay [14,23]. Since the discovery of the green fluorescent protein (GFP) from *Aequorea victoria*, heterologously expressed fluorescent proteins have increasingly been employed as a more useful alternative to *lacZ*, serving as chromogenic reporters in both prokaryotic and eukaryotic systems [24,25]. In stark contrast to *lacZ*, the average ORF size of jellyfish- and coral-derived fluorescent proteins is only 720 bp [26]. These proteins are easily excited using specific wavelengths of light instead of requiring a substrate, allowing emission to be measured in whole cells. Mutated forms have been optimised for a variety of microscopy techniques [27], greater intensity emission [28], altered emission wavelength [29], and performance at higher temperatures [30].

Whilst *lacZ* can be used as a reporter based on the phenotype of growth on lactose as a carbon source, this is rarely the case [31,32,33]. However, other genes have been used in this way, such as the acetamidase-encoding *amdS* gene from *Aspergillus nidulans*, whose product catalyses the hydrolysis of acetamide into the carbon source acetate and the nitrogen source ammonium [18,34,35]. Whilst *amdS* has served as a dominant selectable marker across a range of fungi, growth media using acetamide as the sole nitrogen source also enable the presence and intensity of *amdS* gene expression to function as a surrogate reporter; multiple copies or upregulation of *amdS* leads to stronger growth on acetamide [36,37,38,39,40,41,42,43]. Additional functionality arises through counterselection, a feature unavailable with *lacZ*; growth of *amdS*-expressing organisms is inhibited on media containing fluoroacetamide, as hydrolysis into fluoroacetate inhibits the citric acid cycle [18,44,45,46]. Like the fluorescent proteins, the *amdS* ORF (1647 bp) is substantially smaller than *lacZ*.

*lacZ* is undeniably useful as a reporter but as synthetic biology becomes more complex, there is room to replace this tool with a multifunctional alternative. A useful approach is to combine reporters, such as by transforming an organism with constructs that use multiple promoters to confer multiple surrogate phenotypes such as using a dual fluorescent protein system for *Pseudomonas fluorescens* or a dual luciferase assay designed for *Saccharomyces cerevisiae* [47,48]. The next evolution of this approach is driving multiple reporters from a single promoter using a polycistronic transcript such as a dual fluorescent protein system in *E. coli*, fluorescent protein and *lacZ* metabolic activity in *Pseudomonas putida* and *E. coli*, or *lacZ* and *tet* selection in *P. putida* [32,33,49,50]. However, in eukaryotes, polycistronic transcripts are rare and/or ineffective. A more advanced take on this strategy would be fusing reporters together as a single ORF, an uncommon approach that can enable complex experiments such as GFP fused in-frame to *lacZ* in *Caenorhabditis elegans* or heterologously expressed yeast *URA3* and *HIS3* being fused with mCherry and GFP, respectively, in *E. coli* [31,51].

A powerful, multifunctional reporter is lacking in the basidiomycete yeast *Cryptococcus neoformans*, the World Health Organization’s top-ranked critical priority group species on their Fungal Priority Pathogen list [52]. Whilst *C. neoformans* is inhaled, presenting as a lung infection causing pneumonia-like symptoms, it shows a remarkable tropism for the central nervous system where it causes meningoencephalitis [53,54]. Due to the clinical significance of this species, a variety of robust tools such as a diverse range of fluorophores and markers with useful surrogate phenotypes has already been developed [39,55].

Described here is an investigation into the creation of a fused reporter system for *C. neoformans* that combines the most useful fluorophores with the versatility of the counter-selectable activity of *amdS*. Smaller than *lacZ* but providing multiple surrogate phenotypes for interrogation, the fusion ORF serves as a superior whole-cell assay to traditional systems. Broadly, this combines the qualitative, live-cell capacity of fluorescent protein expression linked to the metabolic activity of a counter-selectable marker. The gene fusion of *amdS* with either mRuby3 or mNeonGreen displays stable expression in *C. neoformans* and works effectively when driven by both constitutive (*TEF1*) and inducible (*GAL7*) promoters. The novel gene fusion has potent applicability as a reporter system in *C. neoformans*, which will allow for a more detailed investigation into in vivo expression of genetic elements.

## 2. Materials and Methods

### 2.1. Media

The bacterial strains used in this study are listed in Appendix A and were grown on LB + 100 μg/mL ampicillin media (1% tryptone, 0.5% yeast extract, 1% sodium chloride). The fungal strains used in this study are listed in Appendix A and were grown on YPD (2% bacteriological peptone, 1% yeast extract, 2% agar, 2% glucose) for 2 days at 30 °C. All strains were archived in 15% glycerol and stored at −80 °C. YNB minimal liquid media was also used (0.45% yeast nitrogen base without amino acids and ammonium sulphate, 2% agar, 2% glucose, 10 mM ammonium sulphate). Supplementation of YNB with 1 mM adenine was also used.

### 2.2. Plasmid Design and Creation

All plasmids generated are listed in Appendix A. All constructs were validated for nucleotide-level accuracy using Sanger sequencing (Australian Genome Research Facility, Brisbane, Australia). Unless otherwise stated, constructs were designed to be built into a plasmid that would enable targeting to Safe Haven 1 (pSDMA57), a location in the genome shown to be a neutral site where constructs can be integrated without causing any known phenotypic consequences. Phusion High Fidelity DNA Polymerase (New England Biolabs, Ipswich, MA, USA) was used to amplify the fragments prior to assembly into pSDMA57 Safe Haven *NEO* resistance gene plasmid backbone; a variant of pBluescriptII SK(-), using NEBuilder (New England Biolabs, USA).

The *amdS* ORF was amplified from a previously generated construct pGWKS53 using primers UQ5391 and UQ5393. A SwaI restriction endonuclease cut site was engineered into UQ5391 to allow for subsequential insertion of additional genetic material upstream, intended to be a promoter sequence. The mRuby3 + *ADE6*_(t)_ fragment was amplified from a previously generated construct pGWKS7 using primers UQ5392 and UQ5394. The mNeonGreen + *ADE6*_(t)_ fragment was amplified from a previously generated construct pGWKS11 using primers UQ5395 and UQ5394. These processes led to the generation of mRuby3 variant pLSPR5 and mNeonGreen variant pLSPR6, respectively.

Into each bicistronic variant, either the *TEF1* or *GAL7* promoter was inserted. The bicistronic reporter plasmids were digested with SwaI restriction endonuclease then combined with H99O amplified genomic DNA of either *TEF1*_(p)_ using UQ5397 and UQ5398 or *GAL7*_(p)_ UQ5399 and UQ5400. Combining pLSPR5 with *TEF1*_(p)_ yielded pLSPR9; with *GAL7*_(p)_ yielded pLSPR7, and pLSPR6 with *TEF1*_(p)_ yielded pLSPR10; with *GAL7*_(p)_ yielded pLSPR8.

A new *amdS* ORF fragment lacking the STOP codon was generated from pGWKS53 using UQ5391 and UQ6087. Combined with either a mRuby3 + *ADE6*_(t)_ fragment (UQ5392 and UQ5394) or a mNeonGreen + *ADE6*_(t)_ fragment (UQ5395 and UQ5394), two new promoter-less ORF Fusion reporter constructs were generated, pLSPR22 and pLSPR23, respectively.

Into each ORF Fusion reporter variant either the *TEF1* or *GAL7* promoter was inserted. Both pLSPR22 and pLSPR23 promoter-less plasmids were digested with SwaI restriction endonuclease which was then combined with H99O *C. neoformans* genomic DNA amplifications of either *TEF1*_(p)_ or *GAL7*_(p)_. *TEF1*_(p)_ was amplified using UQ5397 and UQ5398 then combined with digested ORF fusion mRuby3 or mNeonGreen plasmid to generate pLSPR27 and pLSPR29, respectively. *GAL7*_(p)_ was amplified using UQ5399 and UQ5400, likewise combined with digested ORF fusion mRuby3 or mNeonGreen plasmid to generate pLSPR26 and pLSPR28, respectively.

The “flipped orientation” variants were generated by amplifying either mRuby3 using UQ6088 and UQ6085 or mNeonGreen using UQ5992 and UQ6086. The mRuby3 fragment was combined with *amdS* from pGWKS53, amplified using UQ5994 and UQ5995 primers and *ADE6*_(t)_ from pGWKS7 using UQ5996 and UQ5394, creating pLSPR24. The mNeonGreen fragment was combined with *amdS* from pGWKS53, amplified using UQ5994 and UQ5995 primers and *ADE6*_(t)_ from pGWKS11 using UQ5996 and UQ5394, creating pLSPR25.

Into each promoter-less “flipped” ORF Fusion reporter variant, pLSPR24 and pLSPR25, either the *TEF1* or *GAL7* promoters were inserted. Promoter-less plasmids were digested with SwaI enzyme and then had H99O *C. neoformans* genomic DNA amplifications of either *TEF1*_(p)_ or *GAL7*_(p)_ inserted. *TEF1*_(p)_ was amplified using UQ5397 and UQ6098 or UQ5998, for mRuby3 and mNeonGreen constructs, respectively. *TEF1*_(p)_ was amplified using UQ5397 and UQ5398 was inserted into pLSPR24, generating pLSPR31. *TEF1*_(p)_ was amplified using UQ5397 and UQ5998 was inserted into pLSPR25, generating pLSPR33. *GAL7*_(p)_ was amplified using UQ5399 and UQ6097 or UQ5997 for mRuby3 and mNeonGreen constructs, respectively. *GAL7*_(p)_ was amplified using UQ5399 and UQ6097 was inserted into pLSPR24 to generate pLSPR30. *GAL7*_(p)_ was amplified using UQ5399 and UQ5997 was inserted into pLSPR25 to generate pLSPR32.

### 2.3. Strain Creation

All strains were generated in *C. neoformans* type strain H99O using biolistic transformation, unless stated otherwise, and are listed in Appendix A. All reporter strains were built into the pSDMA57 plasmid backbone for homologous recombination with the Safe Haven region of the *C. neoformans* genome during transformation. Plasmids were linearised using endonuclease PacI prior to biolistic transformation conducted as described by Arras et al. [56] onto sorbitol (Sigma-Aldrich, St. Louis, MO, USA) and 5 µg/mL W7 hydrochloride (Tokyo Chemical Industry, Oxford, UK)—supplemented media.

### 2.4. Southern Blotting

All *C. neoformans* transformants, unless otherwise stated, were validated through Southern blotting. Genomic DNA was extracted from the strains using the CTAB method [57], digested using endonucleases, and then run on a 1% agarose gel overnight at 20 V. The DNA present in the gel was then Southern blotted onto a Hybond-XL membrane (GE HealthCare, Chicago, IL, USA) [58]. Southern hybridisation requires a radioactively labelled DNA probe; the Safe Haven region of the *C. neoformans* genome served this purpose. It was PCR amplified using primers UQ2962 and UQ2963 and radiolabelled using the DECAprime II kit (Thermo Fisher Scientific, Waltham, MA, USA) and dCTP [α-^32^P] (PerkinElmer, Waltham, MA, USA). The blots were hybridised with the probe at 65 °C overnight, then consecutively washed with a 0.1% SDS and 2% SSC buffer before being exposed to a 20 × 25 Exposure Cassette (GE Healthcare) and imaged on a Typhoon FLA 7000 phosphorimager (GE Healthcare).

### 2.5. Spotting Assays

The desired strains were grown in 15 mL YPD liquid as a starter culture overnight. An amount of 1 mL of the starter culture was used to inoculate 50 mL of fresh YPD liquid to an OD_600_ reading of ~0.3. The cultures were grown in a shaking incubator at 30 °C and 250 rpm. Upon reaching an OD_600_ of 1, 1 mL of each cultured strain was harvested and serially diluted 100 μL into 900 μL of dH_2_O four times. Between 3 and 5 μL of each sample was plated as a single drop onto each agar-media plate. The spotted plates were left to incubate at 30 °C for two days prior to having their phenotype recorded and photographed.

### 2.6. Imaging Plates

The spotting assay plates were imaged with two approaches. Conventional photography was conducted at 1/100 s shutter speed, 200 ISO, and F5 aperture using a Nikon AF Micro Nikkor 60 mm f/2.8 D (Nikon, Minato City, Japan) mounted on a Kaiser R1 system (Kaiser Fototechnik, Buchen, Germany). Fluorescence was captured using an Amersham ImageQuant 800. The mNeonGreen strains were excited using 460 nm light and emission was captured using a Cy2 bandpass filter capturing 505–545 nm emission. The mRuby3 strains were excited using 535 nm light and emission was captured using a Cy3 bandpass filter capturing 565–645 nm emission.

### 2.7. Flow Cytometry

Both *GAL7*_(p)_- and *TEF1*_(p)_-driven strains of both reporters were grown overnight as starter cultures in a yeast nitrogen base without amino acid (YNB) media supplemented with either 2% glucose or 2% galactose. A total of 1 mL of each starter culture was inoculated into 50 mL of fresh YNB media, generating three unique treatment exposures: glucose exposure is maintained, galactose exposure is maintained, and glucose exposure is replaced with galactose exposure. For each of the fluorophore strains, 1 mL of each sample was taken immediately following inoculation as well as every hour for 12 h. *C. neoformans* cells were fixed at each hour timepoint using 4% paraformaldehyde solution. The fixed *C. neoformans* reporter strains were run on a Beckman CytoFLEX flow cytometer to measure the relative fluorescence of mutants following induction of *GAL7*_(p)_ expression. The flow cytometry data was analysed using FlowJo (version 10.10.) and compiled using GraphPad Prism 10.

## 3. Results

### 3.1. Conceiving a Multireporter Construct

To initiate the design of a multifunctional reporter in *C. neoformans*, the elements that bring the most value were first identified. Our previous studies of eleven fluorescent protein constructs in *C. neoformans* revealed a range of useful fluorophores across the visible spectrum. Amongst the red proteins, mRuby3 exhibited higher relative fluorescence than mCherry [55]. Blue, yellow, and maroon constructs were less vibrant; however, amongst the four green constructs tested, mNeonGreen and Clover were both equally easy to detect. mRuby3 and mNeonGreen, two fluorophores with nonoverlapping spectra, were selected for further use; the broad range of intensity would enable visual detection of a large range of expression levels.

To enable growth-based assessment of magnitude of expression, the counter-selectable marker *amdS* was included. A previous study has shown this marker to have remarkably high utility through both its ability to confer growth on acetamide as a nitrogen source as well as sensitivity to fluoroacetamide [39]. Just as the expression of *amdS* enables growth on acetamide as the sole nitrogen source, the expression of this gene on fluoroacetamide media inhibits growth, enabling the investigation of repression or “leaky” expression. The two reporters (one of the two chosen fluorophores alongside *amdS*) were joined using a short and flexible linker, employing nonpolar glycine and polar serine residues to generate a structure that could maintain stability in the presence of an aqueous solvent (Gly-Ser-Gly-Gly-Gly-Gly-Ser-Gly) [59,60,61,62,63].

It was a deliberate decision to include two distinct types of reporters in this system: both a fluorophore and a dominant counter-selectable marker. A fluorophore, either mRuby3 or mNeonGreen, behaves as a “nuanced” expression reporter, whereas *amdS* behaves as a “coarse” expression reporter. The degree of expression output can be discreetly measured as a function of fluorescence signal; hence, this reporter is described as nuanced where greater fluorescence is attributed to directly higher gene transcription. Conversely, growth on media supplemented with acetamide and inhibited growth on media supplemented with fluoroacetamide, the dominant counter-selectable marker *amdS* is considerably more difficult to discreetly measure. As such, growth or lack of growth for *amdS*-expressing *C. neoformans* strains functions as a coarse “yes” or “no” reporter for gene transcription. Through combining both nuanced and coarse reporters for gene transcription, a powerful tool for the evaluation of *C. neoformans* expression has been designed.

The *C. neoformans* genome is intron dense [64,65] and, as heterologous constructs lacking introns tend to be very poorly expressed [66,67,68], these important elements were introduced via an intron-rich 3′ untranslated region. Three of the four introns of the 4833 bp phosphoribosylformylglycinamidine synthase-encoding *ADE6* gene are found in its 569 bp 3′ UTR, establishing it as the most intron-rich 3′ UTR in the genome. A derivative of an intron from the phosphoribosylaminoimidazole carboxylase-encoding *ADE2* gene was also transplanted into the *amdS* section of the construct to increase intron density. An 8 bp unique site for the restriction endonuclease SwaI was incorporated immediately 5′ of the construct to enable the insertion of promoter sequences.

Finally, the complete construct was built into a plasmid that targets integration at Safe Haven 1 (Figure 1). The Safe Haven is a location in chromosome 1 of the *C. neoformans* genome previously shown to be a consistent, neutral site where constructs can be integrated without causing any known phenotypic consequences, enabling comparison between strains where different promoters are used in the reporter system [39,55].

To trial the efficacy of the construct the promoter of the constitutively expressed translational elongation factor EF-1 alpha-encoding *TEF1* gene, a native *C. neoformans* promoter with a 499 bp 5′ UTR that adds another intron to the construct was inserted, resulting in a net count of five introns over 4333 bp of the transcript.

### 3.2. Trialling a Bicistronic Transcript in C. neoformans

Whilst polycistronic transcripts are rare in fungi, they do exist [69]. Given the utility this could provide, the design process began by investigating whether expressing our two reporters as a bicistronic transcript rather than a fusion would produce both products or just one. A bicistronic construct consisting of the native *TEF1* promoter driving expression of a transcript containing the *amdS* ORF (terminated with a stop codon) followed by the mRuby3 ORF (also with its own start and stop codon) was generated, validated through Sanger sequencing and biolistically transformed into *C. neoformans*.

The bicistronic reporter strain was indistinguishable from the wild type on YNB with ammonium as the sole nitrogen source but on acetamide, there was a stark difference—the wild type was unable to grow while the strain bearing the new reporter exhibited strong growth equivalent to a control strain expressing just *amdS* (Figure 2). The *amdS* phenotype was further reinforced through the absence of the growth of the reporter strain and the *amdS* control strain on fluoroacetamide-containing media whilst wild type growth remained robust (Figure 2).

When the reporter strains were exposed to 535 nm light, the emission of fluorescence in the 565–645 nm wavelengths of the visible spectrum was observed from the mRuby3 control strain but no others (Figure 2). Together, these data illustrate that the translated product of the first ORF (*amdS*) in the bicistronic reporter was present and conferred a visible phenotype but the translated product of the second ORF (mRuby3) in the reporter was not. Together, these data indicated that only the first ORF in the reporter transcript was being translated and that a bicistronic reporter of this design was not viable.

### 3.3. Fusing the ORFs Enables Dual Reporter Expression

Rather than pursue research avenues that might possibly confer appropriate functionality to the bicistronic construct, a more straightforward approach that was more likely to succeed was to fuse the two reporters together into a single ORF. The construct was modified by precisely removing the stop codon of the *amdS* ORF. The new construct was also generated, validated through Sanger sequencing and biolistically transformed into *C. neoformans*. Contrary to the previously generated bicistronic reporter, once transformed into *C. neoformans*, the strain carrying this fused construct now exhibited both reporter phenotypes (Figure 3A). The strain grew on acetamide as the sole nitrogen source was sensitive to fluoroacetamide and, when exposed to 535 nm light, produced strong fluorescence between 565 and 645 nm.

Following the success of the mRuby3 dual reporter strain, a variant replacing mRuby3 with mNeonGreen was generated and biolistically transformed into *C. neoformans*. Similar to the mRuby3 strain, the *amdS*:mNeonGreen dual reporter grew on acetamide as the sole nitrogen source; was sensitive to fluoroacetamide; and, when exposed to 460 nm light, emitted fluorescence between 505 and 545 nm (Figure 3B).

### 3.4. Superior Gene Orientation for Reporters

Despite the easily observable fluorescence, the emission of the mNeonGreen:*amdS* reporter strain was visibly weaker than that of the mNeonGreen-only controls (Figure 3B). The opposite was true for mRuby3, with the fusion producing stronger fluorescence than the mRuby3 ORF alone. Whether swapping the order of the reporter ORFs would alter the influence on the level of fluorescence was therefore investigated. Additional plasmids and *C. neoformans* strains were generated placing either the mRuby3 or mNeonGreen ORF upstream (5′) of the *amdS* ORF, all under the control of the constitutive *TEF1* promoter. Like before, these new constructs were generated, validated through Sanger sequencing and biolistically transformed into *C. neoformans*. These “flipped” dual reporters were assayed for phenotypes as before (Figure 4).

Fluorescence emission intensity of both mRuby3 and mNeonGreen was equivalent whether or not each was either upstream or downstream of the *amdS* ORF. Growth on acetamide and susceptibility to fluoroacetamide was also unaffected. With this observation, there was no indication that the order of mRuby3/mNeonGreen:*amdS* influenced the efficacy of the final dual reporter design when the expression was driven using the constitutive *TEF1* promoter.

### 3.5. Using the Reporter Fusion to Better Understand Regulated Expression of GAL7_(p)_

Following the successful demonstration of the dual reporter constructs using the constitutive *TEF1* promoter, the system was employed to evaluate the expression of a widely used inducible promoter from *C. neoformans*—the promoter from the galactose-1-phosphate uridylyltransferase gene *GAL7* [70,71]. *GAL7*_(p)_ is known to be repressed in the presence of the preferred carbon source glucose and induced in the presence of pathway substrate galactose. Similar to the *TEF1*_(p)_-driven constructs, the *GAL7*_(p)_-driven constructs were generated, validated through Sanger sequencing and biolistically transformed into *C. neoformans*, where they were also phenotypically assayed (Figure 5). The plasmid maps and Addgene accession details for all fusion reporter constructs are listed in Appendix A.

Under repression conditions (2% glucose), the *GAL7*_(p)_-driven fusion reporter strains exhibited wild-type phenotypes on all media: they grew on NH_4_ as the nitrogen source, did not utilise acetamide as a nitrogen source, were not sensitive to fluoroacetamide, and did not fluoresce (Figure 5).

When grown under mixed repression and inducing conditions (2% glucose + 2% galactose), strains bearing the *GAL7*_(p)_-driven reporters were indistinguishable from the controls on media with NH_4_ as the nitrogen source. However, the simultaneous presence of both repression (2% glucose) and induction (2% galactose) yielded intermediate phenotypes on other media. On acetamide as the sole nitrogen source, the strains could now grow but not as prolifically as strains that had the reporter constitutively expressed with *TEF1*_(p)_. Resistance to fluoroacetamide was weaker but still relatively robust. And now, weak fluorescence could be observed. These phenotypes highlight that the reporter fusions are capable of yielding intermediate phenotypes reflective of altered transcription from the promoter being employed.

Under exclusively inducing conditions (2% galactose), the *GAL7*_(p)_-driven fusion reporter strains resembled constitutive *TEF1*_(p)_ fusion reporter strains when grown on acetamide as the sole nitrogen source; these strains were indistinguishable. Notably, a difference between *TEF1*_(p)_ constitutive expression and *GAL7*_(p)_ inducible expression was apparent on fluoroacetamide, where the *GAL7*_(p)_ reporter strains were slightly less sensitive to fluoroacetamide; that is, this unique readout enabled the observation that *TEF1*_(p)_ transcription is actually stronger than induced *GAL7*_(p)_ transcription. Taking this further, a difference between the *GAL7*_(p)_-*amdS*:fluorophore and *GAL7*_(p)_-fluorophore:*amdS* fusion order was apparent, with the latter exhibiting higher fluoroacetamide sensitivity indicating a greater abundance of the reporter fusion protein. A difference in fluorescence was also observed, with the *GAL7*_(p)_-*amdS*:fluorophore strains exhibiting less fluorescence than their *GAL7*_(p)_- fluorophore:*amdS* fusion counterparts. In this case, the *amdS* component can yield a relatively coarse output on acetamide but is far more nuanced on fluoroacetamide. Despite the expression of *amdS* being a qualitative output for the strength of the respective transcription regulator, it in combination with a fluorophore capable of more quantitative output demonstrates the potential of the fusion reporter system as an analytical tool for *C. neoformans* research. Together, these data highlight the power of how having three simple detectable plate phenotypes combined with an alternative fusion order can give additional regulatory insight.

### 3.6. Flow Cytometry Analysis of GAL7_(p)_ Expression

Even when growing the reporter strains on a plate and visualising an entire colony, the sensitivity of detection of fluorescence from the fusion markers was excellent, with differences based on either the order of reporter fusion or the level of transcription from the promoter easily observable. The varied fluorescence output from the two reporter arrangements offers an interesting opportunity. If an analysis of high fluorescence expression is desired, the fluorophore upstream of the *amdS* ORF should be utilised. If an analysis of a more nuanced output of the promoter is desired, the reporter arrangement with the fluorophore downstream should be selected. An output “lever” such as this allows for subtle system control that offers power to the experimental design in question. To investigate this sensitivity further, flow cytometry was undertaken to evaluate the fluorescence expression of reporter strains more quantitatively.

Due to the fluorophore:*amdS* arrangement demonstrating the strongest phenotypic expression, to detect expression over the largest range, *GAL7*_(p)_-mRuby3:*amdS* and *GAL7*_(p)_-mNeonGreen:*amdS* were selected to investigate the regulated transcription from *GAL7*_(p)_. All strains were grown in a glucose- or galactose-supplemented YNB starter culture overnight before being inoculated into either glucose- or galactose-supplemented YNB media and grown for a further 12 h, during which they were sampled every hour for flow cytometry analysis (Figure 6).

Flow cytometry was successfully used to investigate the fusion reporter constructs as providing a surrogate phenotype for induced *GAL7* promoter transcription. Both reporter strains were able to track the increase in fluorescence following galactose exposure. Mutant strains were exposed to galactose during growth, inducing *GAL7*_(p)_, with both mRuby3 and mNeonGreen fluorescent output being used to track the increase in fluorescence over time, through relative fluorescence and mean fluorescence intensity (MFI).

The application of flow cytometry to monitor the fluorescence output of mRuby3 and mNeonGreen fusion reporter *C. neoformans* strains also exhibited a clear difference in their ease of applicability for in vivo promoter evaluation. When given sufficient maturation time, despite both fluorescent reporters displaying the capability for exhibiting fluorescence as expected, mRuby3 would appear as the superior fluorophore for growth-dependent fluorescence compared to mNeonGreen. The lower differentiation of relative fluorescence as well as overall lower MFI exhibited by mNeonGreen is hypothesised to result from the autofluorescence intrinsic to the study of *C. neoformans* [55]. For maximising growth-dependent fluorescence output, the mRuby3:*amdS* fusion reporter was observed to be the superior arrangement for promoter transcription evaluation in *C. neoformans*.

Whilst both fluorophore reporters have been demonstrated as valuable promoter evaluation tools, the full repertoire of reporters generated in this project could be investigated further using additional fluorescent techniques including microscopy and additional flow cytometry to elucidate nuances in their surrogate expression phenotypes. The degree of fluorescence output from the fusion reporters substantiated the system as worthwhile progress towards robust expression readouts in the research-critical human pathogen *C. neoformans*.

## 4. Discussion

Historically, there have been two methodological approaches for the investigation of gene expression: in vitro and in vivo. Crucial in vitro methodologies include qPCR, western blots, northern blots, and proteomics. In vivo mechanisms, on the other hand, tend to be more organism-specific and/or laborious to implement, such as fluorescent tagging systems or live-cell staining. The expansion of in vivo gene monitoring techniques is particularly attractive for clear phenotypic assaying of gene expression in the fungal priority pathogen *C. neoformans*. The design and implementation of a system described here are hypothesised to be compatible with other fungi as the key components have now been shown to function in the two largest phyla of the fungi kingdom.

The initial reporter construct was modelled after a bacterial bicistronic system. It was hypothesised that, despite similar systems existing in fungal species, their rarity probably meant that our very basic approach to building a bicistronic system would be unsuccessful [69]. Despite this initial lack of bicistronic system success, such a genetic arrangement may prove worth revisiting; perhaps with a focus on engineering the inter-ORF space to be more likely recognised by the ribosome. Optimising the linker sequence to contain the Kozak sequence upstream of the *amdS* ORF may alleviate the issues otherwise encountered in the bicistronic attempts while simultaneously permitting more control over the reporter system [72]. Whilst the uncertainty of success with any of these types of interventions led us to abandon a bicistronic reporter design for now, it opened up an intriguing direction for investigation in the development of new genetic tools for this important pathogen in the future.

Trials of the initial fusion reporter constructs expressed in *C. neoformans* demonstrated that, despite some variance in intensity between the control strains individually expressing mRuby3 or mNeonGreen, the fusion reporters were equivalent in fluorescence output. The phenotype coinciding with *amdS* expression was also observed as expected; the strain bearing the reporter fusion utilised acetamide as the sole nitrogen source and was indistinguishable from the strain individually expressing *amdS*. Fluorescence expression and acetamide utilisation remained consistent when the fluorophore and *amdS* sequence orientation was swapped; when bounded by *TEF1*_(p)_ and *ADE6*_(t)_, no differential transcription output was observed.

Following the success of the fusion transcripts driven by the constitutive *TEF1* promoter, the native *C. neoformans* inducible promoter *GAL7*_(p)_ was similarly investigated. Interestingly, upon spotting assay replication, a comparatively weaker expression was observed by some fusion reporter constructs driven by *GAL7*_(p)_. Despite exhibiting equivalent acetamide utilisation to the other reporter strains, *GAL7*_(p)_-driven *amdS*:mRuby3 and *amdS*:mNeonGreen variants produced noticeably weaker fluorescence emission. Growth on acetamide is a coarse measure of transcription output; thus, the effect from downstream ORF arrangement was more difficult to observe.

Such an observation may be in part due to the negative correlation between the length of the transcript and the integrity of the encoded protein [73,74,75,76]. Impacted by the rate of mRNA degradation and the density of ribosomes, the longer the transcript, the more translation issues may arise. The reduction to fluorescence was not observed for the flipped reporter orientation; when either fluorophore was upstream of the *amdS* ORF, *GAL7*_(p)_-driven reporters appeared equivalent to the *TEF1*_(p)_-driven counterparts. Similarly, no observable reduction in *amdS* functionality via change to acetamide utilisation nor susceptibility to fluoroacetamide was exhibited for the flipped orientation strain either.

The nuance of the fluorescence output is highlighted by the *GAL7*_(p)_ phenotype, demonstrating powerful potential in the fusion reporter system. For a promoter where expression levels are high, having the fluorophore downstream enables easier visualisation of the uppermost levels of transcription. Likewise, for lower levels of expression, the fluorophore located upstream would provide superior fluorescent resolution. This dynamic between a coarse phenotypic output, *amdS*, and a nuanced phenotypic output, mRuby3/mNeonGreen, permits a useful “lever” for the desired investigation of transcription control in an organism. The use of both arrangements of reporter composition, as well as varied analytical approaches such as relative fluorescence and MFI from flow cytometry, can permit a high degree of promoter transcription resolution for future *C. neoformans* research.

For the purpose of further elucidating the control mechanisms influencing the expression of *GAL7*_(p)_ in *C. neoformans*, the fluorophore:*amdS* variant fusion reporter was selected as the arrangement used in the flow cytometry investigation. The two-hour lag observed in the relative fluorescence of mRuby3 emission appears consistent with the fluorescent protein’s known maturation time of 136.5 min [77]. Having been reported as a problem in *C. neoformans* cells previously, the flow cytometry investigation suggested that cells exhibit a degree of autofluorescence that somewhat counteracts the emission wavelengths produced by mNeonGreen strains [55,78,79,80,81]. This limitation was observed only in a growth curve analysis; when more cells are present, especially during the stationary growth period, mNeonGreen fluorescence is easier to distinguish. Similarly, optimisation of fluorescence capturing during flow cytometry may reduce the background signal encountered with mNeonGreen. Tangentially, despite the procedure described in this study, fixing *C. neoformans* cells is not necessary for the usage of the reporter system, which was instead performed as a safety requirement for the equipment used.

Whilst the mNeonGreen fusion reporter should not be overlooked, the mRuby3 variant appears to be the most viable system for future work if it involves flow cytometry. Furthermore, it may be beneficial to continue to evaluate the fusion reporter system described here against staple fluorescent antibody controls utilised in other flow cytometry protocols. This will enable the gauging of relative power and therefore analytical potential the fusion reporter system has in the landscape of fluorescence output. Ultimately, the fusion reporter system has been demonstrated to accurately and successfully output surrogate phenotypes to represent the expression of otherwise obscure transcription drivers by *TEF1*_(p)_ and *GAL7*_(p)_ through both spotting assay and flow cytometry investigation.

Irrespective of the success of the fusion reporter system, the progress made towards a dual-function analysis tool has further iterative potential. This project detailed two recently codon-optimised fluorescent proteins available in the *C. neoformans* background: mRuby3 and mNeonGreen, covering the red and green visible spectrum, respectively. These fluorophores are strong sources of fluorescence but offer limited coverage of the fluorescent repertoire. It may be valuable to extend the fusion reporter construct we have developed to other powerful fluorescent proteins, now made available to *C. neoformans*. Both Citrine and mTurquoise2 fluorophores also were demonstrated to be potent in the *C. neoformans* background and would broaden the spectrum for fusion reporters in the pathogen to aid continued research [55].

Both mRuby3 and mNeonGreen have been successfully adapted into fusion constructs alongside the *amdS* gene as a dominant counter-selectable marker. Despite the *amdS* sequence being codon-optimised for *C. neoformans*, spontaneous loss of function mutation conferring fluoroacetamide resistance is still a possibility. Awareness of this limitation to the *amdS* counter-selectable marker in *C. neoformans* expression is crucial for accurate interpretation in the future use of the described fusion reporter system.

The combination of the mRuby3 and mNeonGreen fluorophores and the *amdS* dominant counter-selectable marker offers a valuable novel tool for evaluating gene expression. The fusion reporter system is capable of both qualitative and quantitative outputs. Both the fluorophores and *amdS* exhibit an obvious visual phenotype: the fluorophores either fluoresce if transcription is active or not if transcription is inactive. Likewise, *amdS* either is capable of growth on acetamide and dies on fluoroacetamide if active or fails to grow on acetamide and lives on fluoroacetamide if inactive. The fusion reporter is a highly successful qualitative output for the presence or absence of gene transcription. Additionally, both the fluorophores and *amdS* can be used more quantitatively to evaluate gene expression: the intensity of fluorescence can be discreetly measured and attributed to the degree of gene expression occurring at that *C. neoformans* locus under the specific growth conditions. Similarly, the extent to which a strain is capable of growth on acetamide could function as a quantitative measure linked to *amdS* expression, albeit a slightly less powerful one. The fusion reporter system described here has potential as both a powerful qualitative and quantitative measure of gene expression in *C. neoformans*.

Additionally, the fusion reporter system is applicable to evaluating any expression-driving sequence in the *C. neoformans* genome. As a result of incorporating the SwaI endonuclease restriction site adjacent to the beginning of the reporter ORF, virtually any sequence capable of driving transcription could be inserted into the reporter system and biolistically transformed into the *C. neoformans* genome for powerful phenotyping. Evaluation of other promoter sequences could also be conducted using this fusion reporter system. Whilst the selection of promoters currently used in the study of *C. neoformans* is somewhat limited (*ACT1*_(p)_, *TEF1*_(p)_, *GPD*_(p)_, *GAL7*_(p)_, and *CTR4*_(p)_), any feasible driver of transcription could be investigated, leading to the utilisation of this fusion reporter system as a means to evaluate novel promoter creations, perhaps resulting from random mutagenesis or sequence-bashing. Whilst the focus of this study was facilitated by investigating two native *C. neoformans* promoters, a similar yet equally important investigation of native terminator expression could be highly relevant. A “drop-in” endonuclease site for terminator sequences similar to what was engineered in this study for promoter sequences could be easily accomplished, providing an even more powerful tool for the interrogation of *C. neoformans* expression elements.

Furthermore, as maintaining pathogenicity is a key component of *C. neoformans* study, ensuring that all constructs introduced do not influence virulence is paramount. Despite each component of the fusion reporter system having been previously shown to not affect virulence when integrated into the Safe Haven site, validation that the entire fusion reporter system behaves similarly would prove to be valuable and should be undertaken in the immediate future [4,5,29].

The fusion reporters described here have been engineered and optimised for function in the pathogen *C. neoformans* but there is an opportunity to develop a derivative system for other relevant pathogenic fungal species such as *Candida auris*, *Candida albicans*, or *Aspergillus fumigatus*; all ranked as critical in the WHO’s fungal priority pathogens list.

## Figures and Tables

**Figure 1 jof-10-00567-f001:**
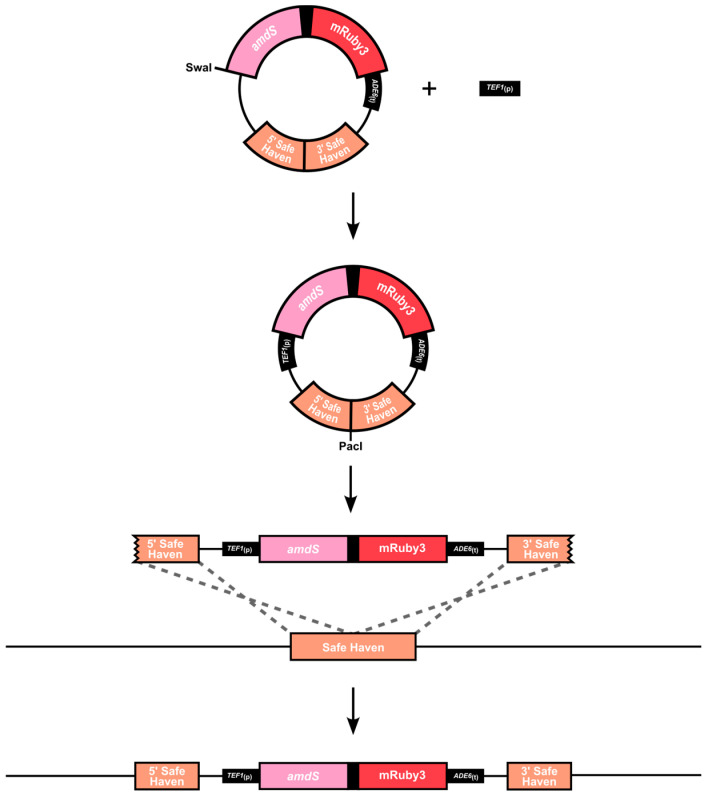
Schematic overview of the bicistronic reporter construct integrating into *C. neoformans* Safe Haven locus. Built into the pSDMA57 plasmid backbone, engineered to be linearised using PacI endonuclease. The *amdS*-mRuby3 bicistronic reporter construct utilises the 5′ and 3′ flanking regions of the Safe Haven as regions of homologous recombination for integration. Expression is driven by the constitutive promoter of *TEF1*, which was inserted using the SwaI endonuclease site. The neomycin resistance gene (*NEO*) present in the plasmid backbone was used as the selectable marker to identify successful gene deletion mutants.

**Figure 2 jof-10-00567-f002:**
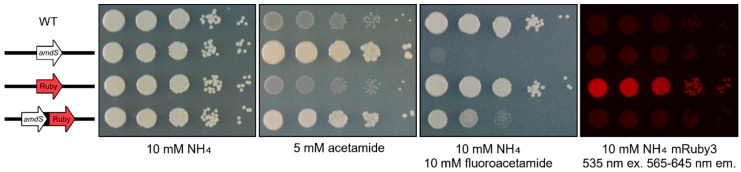
Only the first of two ORFs is expressed in the bicistronic reporter construct. Spotting assay of *amdS*:mRuby3 bicistronic strain on 2% glucose YNB agar media plates supplemented as depicted. Strains individually expressing *amdS* or mRuby3 were used as controls. mRuby3 was excited using 535 nm light, capturing fluorescence between 565–645 nm. The wild-type control is H99O. Plates were photographed following 72 h of incubation at 30 °C.

**Figure 3 jof-10-00567-f003:**
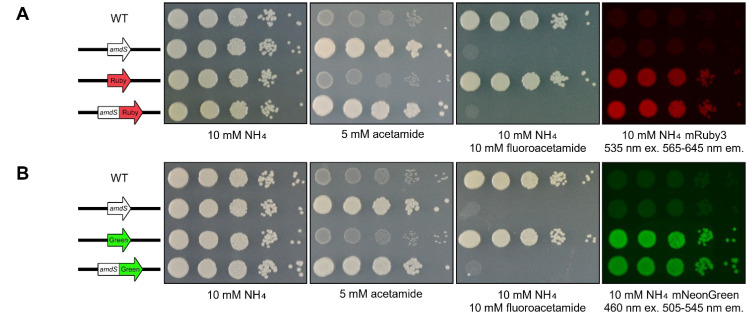
Fusing two ORFs produces reporters with multiple useful phenotypes. (**A**) Spotting assay of *amdS*:mRuby3 fusion strain on 2% glucose YNB agar media plates supplemented as depicted. mRuby3 was excited using 535 nm light, capturing fluorescence between 565 and 645 nm. (**B**) Spotting assay of *amdS*:mNeonGreen fusion strain on 2% glucose YNB agar media plates supplemented as depicted. mNeonGreen was excited using 460 nm light, capturing fluorescence between 505 and 545 nm. Strains individually expressing *amdS*, mRuby3 or mNeonGreen were used as controls. The wild-type control is H99O. Plates were photographed following 72 h of incubation at 30 °C.

**Figure 4 jof-10-00567-f004:**
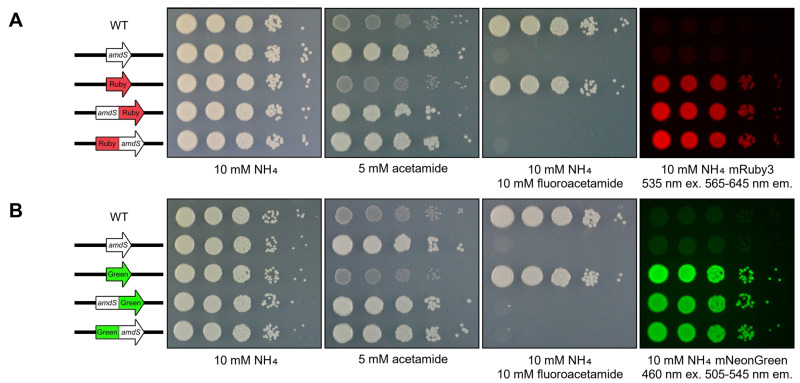
Order of ORF fusion does not affect phenotypic output for the fusion reporter when constitutively expressed using *TEF1*_(p)_. (**A**) Spotting assay of mRuby3:*amdS* fusion strain on 2% glucose YNB agar media plates supplemented as depicted. mRuby3 was excited using 535 nm light, capturing fluorescence between 565 and 645 nm. (**B**) Spotting assay of mNeonGreen:*amdS* fusion strain on 2% glucose YNB agar media plates supplemented as depicted. mNeonGreen was excited using 460 nm light, capturing fluorescence between 505 and 545 nm. Strains individually expressing *amdS*, mRuby3 or mNeonGreen were used as controls. The wild-type control is H99O. Plates were photographed following 72 h of incubation at 30 °C.

**Figure 5 jof-10-00567-f005:**
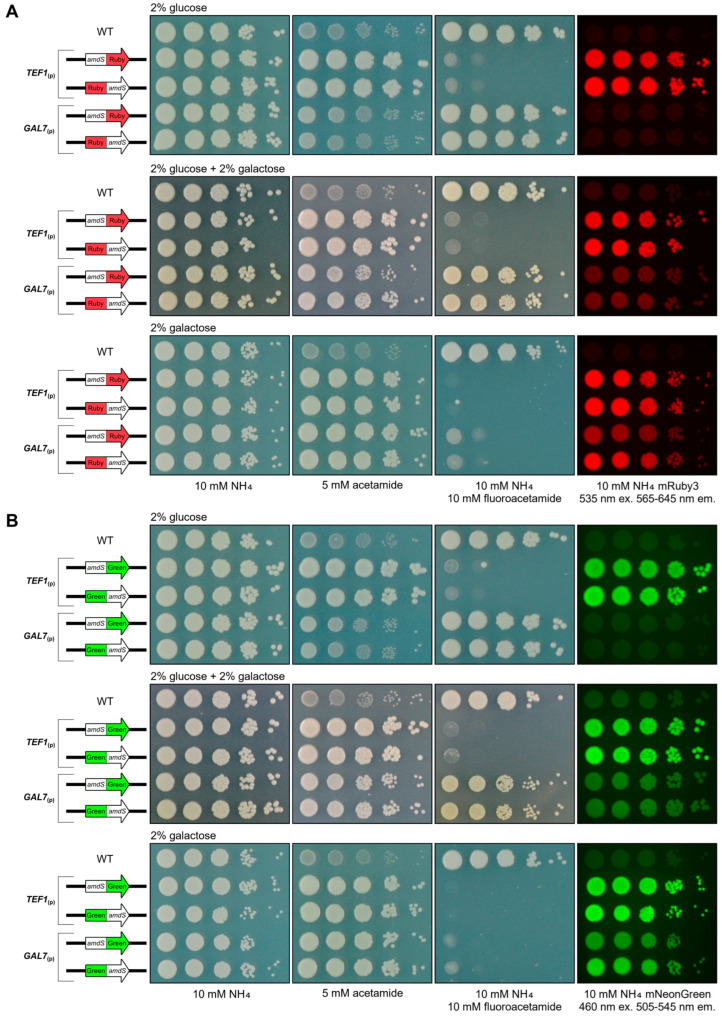
Order of ORF fusion affects fluorescence phenotype for the fusion reporter when expressed using inducible *GAL7*_(p)_. Both orientations of the mRuby3 and mNeonGreen dual fusion reporters driven by *GAL7* promoter plated on supplemented YNB agar. Controls are wild-type H99O and the four reporter variants under the control of the constitutive *TEF1* promoter. Plates were photographed following 72 h of incubation at 30 °C. *C. neoformans* reporter strains in a spotting assay on 2% glucose, 2% glucose + 2% galactose, or 2% galactose. Strains were grown on media containing the indicated supplements. (**A**) mRuby3 was excited using 535 nm light, capturing fluorescence between 565 and 645 nm. (**B**) mNeonGreen was excited using 460 nm light, capturing fluorescence between 505 and 545 nm.

**Figure 6 jof-10-00567-f006:**
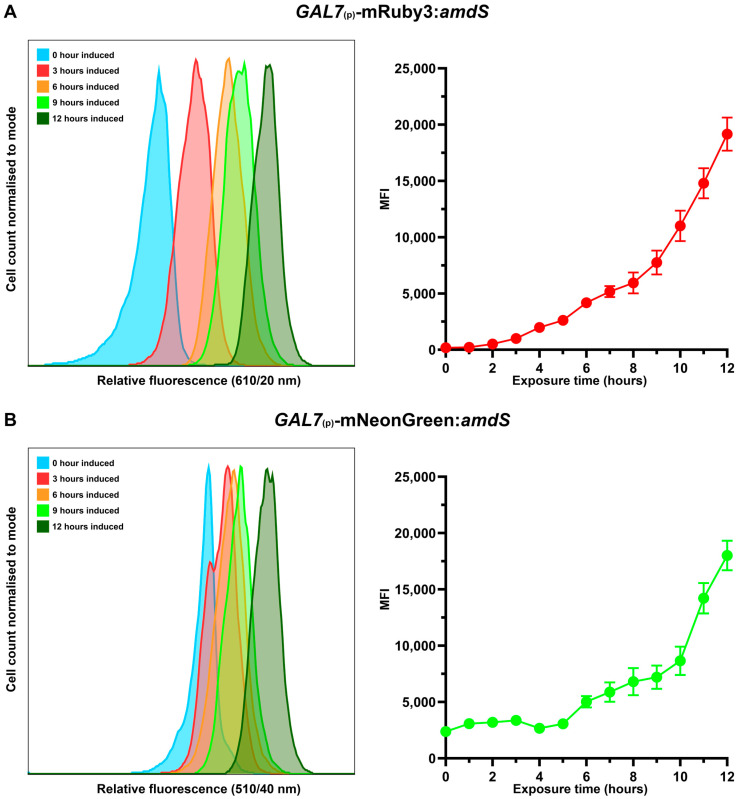
Fluorescence of fusion reporters evaluates inducible promoter expression. *GAL7*_(p)_-fluorophore:*amdS* fusion reporter strains were exposed to galactose-supplemented liquid YNB media following growth on 2% glucose. *C. neoformans* cells were incubated over 12 h and sampled at hourly time points for flow cytometry analysis. Relative fluorescence was calculated using maximal *TEF1*_(p)_-fluorophore:*amdS* strains run through CytoFlexS flow cytometry machine. Cell count (histogram) was standardised (20,000 cells) and relative fluorescence was recorded and displayed for each time point. Mean fluorescent intensity (MFI) was calculated using FlowJo analysis software. Data were collected across five biological replicates. Gating methodology is detailed in Appendix A. (**A**) mRuby3 fluorescence was captured with a 610 ± 10 nm bandpass filter. (**B**) mNeonGreen fluorescence was captured with a 510 ± 20 nm bandpass filter.

## Data Availability

All data described in this study are presented in the manuscript and publicly available. Plasmids included in this publication are available on Addgene (https://www.addgene.org/) under IDs 217307-217316, detailed in the Appendix A. *C. neoformans* strains created and used in this publication are available from the Fungal Pathogenesis Laboratory upon request.

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
