# Peer review of "A Chimeric ORF Fusion Phenotypic Reporter for Cryptococcus neoformans"

_jof, 2024, doi:10.3390/jof10080567_

Round 1
Reviewer 1 Report
This is a report on construction and utilization of a dual reporter system in fungal pathogen Cryptococcus neoformans. The design is reasonable and results are conclusive and clearly presented. There is no major weakness in the report.
The study is overall well executed and clearly presented. The reporter system will be a valuable tool for improving research on Cryptococcus neoformans. The only minor comment I have is that authors may want to show their plasmid design strategy in a figure to illustrate the detail of their design. Also it may be necessary to list the name and/or ID number of the plasmids they deposited to Addgene in the data availability statement.
Author Response
Comments 1: The only minor comment I have is that authors may want to show their plasmid design strategy in a figure to illustrate the detail of their design.
Response 1: Thank you for this comment. To address this comment two additional figures have been created, one in the main body text to display the plasmid map and integration into the Safe Haven region, and another in the Supplementary materials which schematically represents all the plasmids designed, created and made available through Addgene. See the images provided below:
Figure 1: Schematic overview of the bicistronic reporter construct integrating into C. neoformans Safe Haven locus. Built into the pSDMA57 plasmid backbone, engineered to be linearised using PacI endonuclease. The amdS-mRuby3 bicistronic reporter construct utilises the 5’ and 3’ flanking regions of the Safe Haven as regions of homologous recombination for integration. Expression is driven by the constitutive promoter of TEF1, which was inserted using the SwaI endonuclease site. The neomycin resistance gene (NEO) present in the plasmid backbone (not shown) was used as the selectable marker to identify successful gene deletion mutants.
Supplementary Figure 1: Schematics of finalised plasmid maps for fusion reporter constructs. All listed constructs are available on Addgene database. A) Promoter-less fusion reporter constructs for variant promoter combinations, both initial and flipped orientations. B) Fusion reporters driven by GAL7(p) for both initial and flipped orientations. C) Fusion reporters driven by TEF1(p) for both initial and flipped orientations.
Comments 2: Also it may be necessary to list the name and/or ID number of the plasmids they deposited to Addgene in the data availability statement.
Response 2: Thank you for this comment. The plasmid IDs for Addgene access are now listed in Supplementary Materials via the above figure, as well they will be added to the data availability statement too.
Reviewer 2 Report
In Summary:
In the paper “A Chimeric ORF Fusion Phenotypic Reporter for Fungi: Utility in a WHO Priority Pathogen” the authors designed and tested the reporter intending to provide an “even more powerful tool” for the interrogation of C. neoformans expression elements.
My concerns/requests:
1. Please change the title of your paper: the “Utility in a WHO Priority Pathogen” is irrelevant to this highly experimental research.
2. Please make a workflow of your cloning methodology, based on materials&methods paragraph: it isn't easy to understand the rationale for the procedure steps. The procedure should be clear so that other scientists can reproduce it. The supplementary figure would be fine.
3. “Data was collected across two biological replicates”. Two biological replicates are not enough for in vitro research. Please add a minimum of 3, ideally 5 replicates for all experiments that you submitted.
4. The assay results should be quantitative: the “exhibiting higher fluoroacetamide sensitivity” is not a quantitative result. Please calculate statistical significance.
see above.
Author Response
Comments 1: Please change the title of your paper: the “Utility in a WHO Priority Pathogen” is irrelevant to this highly experimental research.
|
Response 1: Thank you for your advice for the title of the manuscript. The title has now been updated to: “A Chimeric ORF Fusion Phenotypic Reporter for Cryptococcus neoformans”.
|
Comments 2: Please make a workflow of your cloning methodology, based on materials&methods paragraph: it isn't easy to understand the rationale for the procedure steps. The procedure should be clear so that other scientists can reproduce it. The supplementary figure would be fine.
|
Response 2: Thank you for the feedback. In order to respond to this comment, several changes have been made to the manuscript in multiple sections. Please see below for details.
Figure 1: Schematic overview of the bicistronic reporter construct integrating into C. neoformans Safe Haven locus. Built into the pSDMA57 plasmid backbone, engineered to be linearised using PacI endonuclease. The amdS-mRuby3 bicistronic reporter construct utilises the 5’ and 3’ flanking regions of the Safe Haven as regions of homologous recombination for integration. Expression is driven by the constitutive promoter of TEF1, which was inserted using the SwaI endonuclease site. The neomycin resistance gene (NEO) present in the plasmid backbone (not shown) was used as the selectable marker to identify successful gene deletion mutants.
Figure order in text body and figure captions has been updated accordingly throughout.
Additionally, to bolster the rationale of the plasmid design, a paragraph was added to the Results at line 247-258: “It was a deliberate decision to include two distinct types of reporters in this system: both a fluorophore and a dominant counter-selectable marker. A fluorophore, either mRuby3 or mNeonGreen, behaves as a ‘nuanced’ expression reporter whereas amdS behaves as a ‘coarse’ expression reporter. The degree of expression output can be discreetly measured as a function of fluorescence signal; hence this reporter is described as nuanced where greater fluorescence is attributed to directly higher gene transcription. Conversely, growth on media supplemented with acetamide and inhibited growth on media supplemented with fluoroacetamide, the dominant counter-selectable marker amdS is considerably more difficult to discreetly measure. As such, growth or lack of growth for amdS-expressing C. neoformans strains functions as a coarse ‘yes’ or ‘no’ reporter for gene transcription. Through combining both nuanced and coarse reporters for gene transcription, a powerful tool for evaluation of C. neoformans expression has been designed.”
The sentence on line 290-293 was changed from: “A bicistronic construct consisting of the native TEF1 promoter driving expression of a transcript containing the amdS ORF (terminated with a stop codon) followed by the mRuby3 ORF (also with its own start and stop codon) was biolistically transformed into C. neoformans.” To: “A bicistronic construct consisting of the native TEF1 promoter driving expression of a transcript containing the amdS ORF (terminated with a stop codon) followed by the mRuby3 ORF (also with its own start and stop codon) was generated, validated through Sanger sequencing and biolistically transformed into C. neoformans.”
A sentence was added on line 319 to clarify workflow: “The construct was modified by precisely removing the stop codon of the amdS ORF. The new construct was also generated, validated through Sanger sequencing and biolistically transformed into C. neoformans. Contrary to the previously generated bicistronic reporter, once transformed into C. neoformans, the strain carrying this fused construct now exhibited both reporter phenotypes (Figure 3A).”
A sentence was added to line 348 now reading: “Additional plasmids and C. neoformans strains were generated placing either the mRuby3 or mNeonGreen ORF upstream (5’) of the amdS ORF, all under the control of the TEF1 promoter. Like before, these new constructs were generated, validated through Sanger sequencing and biolistically transformed into C. neoformans. These ‘flipped’ dual reporters were assayed for phenotypes as before (Figure 4).”
A new sentence was added to line 372: “GAL7(p) is known to be repressed in the presence of the preferred carbon source glucose and induced in the presence of pathway substrate galactose. Similar to the TEF1(p)-driven constructs, the GAL7(p)-driven constructs were generated, validated through Sanger sequencing and biolistically transformed into C. neoformans where they were also phenotypically assayed (Figure 5). Plasmid maps and Addgene accession details for all fusion reporter constructs are listed in Supplementary Figure 1.”
These changes are aimed at clarifying the workflow of this research.
A typo was corrected on line 443: “Both reporter strains were able to track the increase in fluorescence following galactose exposure promoter.” The sentence on line 443 now reads: “Both reporter strains were able to track the increase in fluorescence following galactose exposure.”
A paragraph has been added on line 568-582: “Awareness of this limitation to the amdS counter-selectable marker in C. neoformans expression is crucial for accurate interpretation in future use of the described fusion reporter system.
The combination of the mRuby3 and mNeonGreen fluorophores and the amdS dominant counter-selectable marker offers a valuable novel tool for evaluating gene expression. The fusion reporter system is capable of both qualitative and quantitative outputs. Both the fluorophores and amdS exhibit an obvious visual phenotype: the fluorophores either fluoresce if transcription is active or not if transcription is inactive. Likewise, amdS either is capable of growth on acetamide and dies on fluoroacetamide if active or fails to grow on acetamide and lives on fluoroacetamide if inactive. The fusion reporter is a highly successful qualitative output for the presence or absence of gene transcription. Additionally, both the fluorophores and amdS can be used more quantitatively to evaluate gene expression: the intensity of fluorescence can be discreetly measured and attributed to the degree of gene expression occurring at that C. neoformans locus under the specific growth conditions. Similarly, the extent to which a strain is capable of growth on acetamide could function as a quantitative measure linked to amdS expression, albeit a slightly less powerful one. The fusion reporter system described here has potential as both a powerful qualitative and quantitative measure of gene expression in C. neoformans.
Additionally, the fusion reporter system is applicable to evaluating any expression-driving sequence in the C. neoformans genome.” 2 sentences were added on line 587-593 “Additionally, the fusion reporter system described is applicable to evaluating any expression-driving sequence in the C. neoformans genome. As a result of incorporating the SwaI endonuclease restriction site just upstream to the beginning of the first ORF start site, virtually any sequence capable of driving transcription could be ligated into the reporter system and biolistically transformed into the C. neoformans genome for powerful phenotyping. Evaluation of other promoter sequences could also be conducted using this fusion reporter system. Whilst the selection of promoters currently used in the study of C. neoformans is somewhat limited (ACT1(p), TEF1(p), GPD(p), GAL7(p) and CTR4(p)), any feasible driver of transcription could be investigated, leading to the utilisation of this fusion reporter system as a means to evaluate novel promoter creations, perhaps resulting from random mutagenesis or sequence-bashing. Whilst the focus of this study was facilitated by investigating two native C. neoformans promoters, a similar yet equally important investigation of native terminator expression could be highly relevant. A ‘drop-in’ endonuclease site for terminator sequences similar to what was engineered in this study for promoter sequences could be easily accomplished, providing an even more powerful tool for the interrogation of C. neoformans expression elements.”
2 sentences have been added on line 598-603: “A ‘drop-in’ endonuclease site for terminator sequences similar to what was engineered in this study for promoter sequences could be easily accomplished, providing an even more powerful tool for the interrogation of C. neoformans expression elements.
Furthermore, as maintaining pathogenicity is a key component of C. neoformans study, ensuring that all constructs introduced do not influence virulence is paramount. Despite each component of the fusion reporter system has been previously shown to not affect virulence when integrated into the Safe Haven site, validation that the entire fusion reporter system behaves similarly would prove to be valuable, and should be undertaken in the immediate future [4,5,29].
The fusion reporters described here have been engineered and optimised for function in the pathogen C. neoformans, but there is opportunity in developing a derivative system for other relevant pathogenic fungal species such as Candida auris, Candida albicans or Aspergillus fumigatus; all ranked as critical in the WHO’s fungal priority pathogens list.”
An additional Supplementary Figure has been designed to further clarify the workflow of generating the fusion reporter constructs: Supplementary Figure 1: Schematics of finalised plasmid maps for fusion reporter constructs. All listed constructs are available on Addgene database. A) Promoter-less fusion reporter constructs for variant promoter combinations, both initial and flipped orientations. B) Fusion reporters driven by GAL7(p) for both initial and flipped orientations. C) Fusion reporters driven by TEF1(p) for both initial and flipped orientations.
Comments 3: Data was collected across two biological replicates”. Two biological replicates are not enough for in vitro research. Please add a minimum of 3, ideally 5 replicates for all experiments that you submitted.
Response 3: Thank you for pointing out this problem with this procedure. Flow cytometry work was repeated to feature five biological replicates. Figure 6 was updated and presented below:
Figure 6. Fluorescence of fusion reporters evaluates inducible promoter expression. GAL7(p)-fluorophore:amdS fusion reporter strains were exposed to galactose-supplemented liquid YNB media following growth on 2% glucose. C. neoformans cells were incubated over 12 hours and sampled at hourly timepoints for flow cytometry analysis. Relative fluorescence was calculated using maximal TEF1(p)-fluorophore:amdS strains run through CytoFlexS flow cytometry machine. Cell count (histogram) was standardised (20,000 cells) and relative fluorescence was recorded and displayed for each timepoint. Mean fluorescent intensity (MFI) was calculated using FlowJo analysis software. Data was collected across five biological replicates. Gating methodology is detailed in Supplementary Figure 2. (A) mRuby3 fluorescence was captured with a 610±10 nm bandpass filter. (B) mNeonGreen fluorescence was captured with a 510±20 nm bandpass filter.
Comments 4: The assay results should be quantitative: the “exhibiting higher fluoroacetamide sensitivity” is not a quantitative result. Please calculate statistical significance.
Response 4: Thank you for this comment. To clarify that the expression of amdS is intended to be used as a qualitative not quantitative measure of transcription output, 2 sentences have been added to the relevant section of the manuscript: The section from line 402-417 has been changed from: “Under exclusively inducing conditions (2% galactose), the GAL7(p)-driven fusion reporter strains resembled constitutive TEF1(p) fusion reporter strains when grown on acetamide as the sole nitrogen source; these strains were indistinguishable. Notably, a difference between TEF1(p) constitutive expression and GAL7(p) inducible expression was apparent on fluoroacetamide, where the GAL7(p) reporter strains were slightly less sensitive to fluoroacetamide; that is, this unique readout enabled the observation that TEF1(p) transcription is actually stronger than induced GAL7(p) transcription. Taking this further, a difference between the GAL7(p)-amdS:fluorophore and GAL7(p)-fluorophore:amdS fusion order was apparent, with the latter exhibiting higher fluoroacetamide sensitivity indicating a greater abundance of the reporter fusion protein. A difference in fluorescence was also observed, with the GAL7(p)-amdS:fluorophore strains exhibiting less fluorescence than their GAL7(p)- fluorophore:amdS fusion counterparts. Together, these data highlight the power of how having three simple detectable plate phenotypes combined with alternative fusion order can give more insight into the regulation occurring. In this case, the amdS component can yield a relatively coarse output on acetamide but is far more nuanced on fluoroacetamide.” To: “Under exclusively inducing conditions (2% galactose), the GAL7(p)-driven fusion reporter strains resembled constitutive TEF1(p) fusion reporter strains when grown on acetamide as the sole nitrogen source; these strains were indistinguishable. Notably, a difference between TEF1(p) constitutive expression and GAL7(p) inducible expression was apparent on fluoroacetamide, where the GAL7(p) reporter strains were slightly less sensitive to fluoroacetamide; that is, this unique readout enabled the observation that TEF1(p) transcription is actually stronger than induced GAL7(p) transcription. Taking this further, a difference between the GAL7(p)-amdS:fluorophore and GAL7(p)-fluorophore:amdS fusion order was apparent, with the latter exhibiting higher fluoroacetamide sensitivity indicating a greater abundance of the reporter fusion protein. A difference in fluorescence was also observed, with the GAL7(p)-amdS:fluorophore strains exhibiting less fluorescence than their GAL7(p)- fluorophore:amdS fusion counterparts. In this case, the amdS component can yield a relatively coarse output on acetamide but is far more nuanced on fluoroacetamide. Despite the expression of amdS being a qualitative output for the strength of the respective transcription regulator, it in combination with a fluorophore capable of more quantitative output demonstrate the potential of the fusion reporter system as an analytical tool for C. neoformans research. Together, these data highlight the power of how having three simple detectable plate phenotypes combined with alternative fusion order can give additional regulatory insight.” |
Round 2
Reviewer 2 Report
I am fine with accepting this version (v2).
I am fine with accepting this version (v2).